# Comparison of the Structure, Physicochemical Properties, and Impact on Intestinal Flora of Processed and Unprocessed *Polygonum multiflorum* Starch

**DOI:** 10.3390/foods14091578

**Published:** 2025-04-30

**Authors:** Guiya Yang, Ying Wang, Yuying Hu, Yue Liu, Quan Li, Shuangcheng Ma

**Affiliations:** 1School of Chinese Materia Medica, Beijing University of Chinese Medicine, Beijing 102488, China; yangguiya_310@163.com; 2State Key Laboratory of Drug Regulatory Science, National Institutes for Food and Drug Control, Beijing 102629, China; wangying17@nifdc.org.cn (Y.W.); hyy18853812737@163.com (Y.H.); 3Tianjin Key Laboratory of Therapeutic Substance of Traditional Chinese Medicine, School of Chinese Materia Medica, Tianjin University of Traditional Chinese Medicine, Tianjin 301617, China; quanli85@tjutcm.edu.cn; 4Chinese Pharmacopoeia Commission, Beijing 100061, China

**Keywords:** *Polygonum multiflorum* Thunb, starch, structure, physicochemical properties, intestinal flora

## Abstract

*Polygonum multiflorum* Thunb. (PM) is a starch-rich medicinal herb, but research on the changes in the structure and physical properties of the starch upon processing remains elusive. Herein, the structures and physicochemical properties, particularly the impact on intestinal flora of raw PM starch and processed *Polygonum multiflorum* (PMP) starch, were systematically characterized and compared. XRD and FT-IR results showed that the crystalline structure of PMP starch was disrupted, with the increase in its short-range ordering. Morphological analysis revealed that the size of PMP starch granules increased with the appearance of aggregation. Significant differences in swelling power and solubility were observed, wherein PM starch has a higher swelling power, while its solubility is lower than that of PMP starch. The PM starch also has higher thermal stability. Interestingly, the resistant starch (RS) content in PMP starch was higher, as shown by the in vitro digestibility tests, which is associated with enhanced bioactivity. Moreover, gut microbiota analysis in mice indicated that PMP starch promoted gut health by regulating specific bacterial families. Our current study has offered full insights into the changes of PM starch upon processing, laying a solid foundation for further developing PM starch-derived functional food products.

## 1. Introduction

Starch, primarily found in plants, consists of amylose and amylopectin [1]. Amylose is composed of glucose units linked by α-1,4 glucoside bonds, forming a linear structure with a molecular weight ranging from 1 × 10^5^ to 1 × 10^6^ Da. Amylopectin, on the other hand, consists of numerous smaller amylose molecules connected by α-1,6 glucoside bonds to form branched structures, with a molecular weight ranging from 1 × 10^6^ to 1 × 10^7^ Da [2]. Naturally occurring starch is granular, with a microstructure containing both ordered crystalline regions and disordered amorphous regions. Gelatinization is a process where starch transforms from ordered semicrystalline granules to an amorphous state. The gelation process typically involves the expansion of particles through water absorption, the unwinding of the double helix, the melting of crystals, the breakdown of particle morphology, and the filtration of amylose. The extent of starch gelation is influenced by factors such as moisture content, temperature, heating time, and the presence or absence of shear force [3]. Moreover, further continuous processing of gelatinized starch can destroy its short-range molecular order, increasing the distance between dispersed starch chains. However, the mechanism behind this transformation of short-range molecular order in gelatinized starch remains unclear. Additionally, when lipids and proteins are present in the system, leached amylose can interact with them, forming amylose–lipid/protein complexes [4]. Several studies have shown that starch gelatinized after processing Chinese medicinal herbs differs significantly from its raw form [5,6,7,8]. Radix Aconiti Lateralis Preparata (Fuzi), a potent toxic Chinese medicine, is primarily used in clinical practice after processing. Yang et al. [8] observed that the starch in Fuzi became gelatinized during processing, undergoing repeated cycles of gelatinization and aging, which destroyed its original crystalline structure. While traditional Chinese medicine (TCM) typically contains a significant amount of starch, research on starch in TCM remains limited.

Resistant starch (RS) is a type of dietary fiber that cannot be digested or absorbed in the small intestine of healthy individuals. However, it can be fermented or partially metabolized by bacteria in the large intestine to produce short-chain fatty acids, lactic acid, and a small amount of gas [9]. In recent years, research on resistant starch has garnered significant attention. Zhang et al. [10] prepared RS3 from Canna edulis native starch and demonstrated its ability to significantly alleviate dyslipidemia in hyperlipidemic mice. Previous studies using a high-fat diet-induced hyperlipidemic mouse model have shown that RS derived from yam and winged yam exhibits anti-hyperlipidemic effects [11,12]. Furthermore, RS has been found to increase insulin sensitivity [13], reduce inflammation markers [14,15], lower blood urea nitrogen and serum creatinine levels in kidney disease patients [16], and promote gut health in healthy adults [17]. As a part of the daily diet, consuming an appropriate amount of RS may bring about potential health benefits such as regulating the gut microbiota, controlling body weight, and regulating blood glucose levels. Thus, RS can play a role in disease prevention and treatment through dietary fortification.

*Polygonum multiflorum* Thunb. (PM) is a representative medicinal herb that exhibits different therapeutic effects in its crude and processed forms [18]. Processed *Polygonum multiflorum* (PMP) is more commonly used in clinical practice, primarily due to its tonic and anti-aging properties, whereas PM is typically employed for detoxification, promoting intestinal moisture, and easing bowel movements. Our research team has conducted extensive preliminary investigations into its chemical composition, chemical analysis, toxicity, and processing of PM during the past decade [19]. Polysaccharides are among the primary active constituents of *Polygonum multiflorum*. Early studies by our group focused on the raw polysaccharides of both PM and PMP, with activity closely related to factors such as molecular weight, monosaccharide composition, and glucoside linkage. Importantly, research on the polysaccharides before and after processing *Polygonum multiflorum* has grown in recent years, aiming to better clarify its processing mechanisms [20]. However, starch, a typical edible polysaccharide with content in PM that can be as high as 40%, has been minimally studied in PM, with only two studies providing basic structural insights [21,22]. This implies that despite the abundant starch in *Polygonum multiflorum*, it remains a relatively underexplored area within its research framework. Unlike other well-studied bioactive substances, in-depth starch research may uncover new properties and functions of *Polygonum multiflorum*, opening up a new research perspective. From the functional food perspective, due to the high starch content, studying it can facilitate the development of novel *Polygonum multiflorum*-based functional foods. By thoroughly grasping the structural, physicochemical, and functional characteristics of the starch in Polygonum multiflorum, we are better positioned to skillfully incorporate it into food formulas. This will result in the creation of products featuring unique nutritional value and health-promoting functions.

Hence, in this study, the structures and physicochemical properties, particularly the impact on intestinal flora of raw PM starch and processed *Polygonum multiflorum* (PMP) starch, were systematically characterized and compared. XRD and FT-IR results showed that the crystalline structure of PMP starch was disrupted, with the increase in its short-range ordering. Morphological analysis with SEM revealed that the size of PMP starch granules increased with the appearance of aggregation. Significant differences in swelling power and solubility were observed, wherein PM starch has a higher swelling power, while its solubility is lower than that of PMP starch. The PM starch also has higher thermal stability. Interestingly, the resistant starch (RS) content in PMP starch was higher, as shown by the in vitro digestibility tests, and is associated with enhanced bioactivity. Moreover, gut microbiota analysis in mice indicated that PMP starch promoted gut health by regulating specific bacterial families. Our current study has offered full insights into the changes of PM starch upon processing, laying a solid foundation for further developing PM starch-derived functional food products.

## 2. Materials and Methods

### 2.1. Materials

The raw and processed *Polygonum multiflorum* Thunb (PM) were obtained from a traditional Chinese Medicine (TCM) market (Anguo, Hebei, China). Total Starch Assay Kit and Amylose Assay Kit were provided by Beijing Solarbio Biochemistry Co., Ltd. (Beijing, China). Glucose Assay Kit (GOPOD format) was provided by Beijing Biosharp Biochemistry Co., Ltd. (Beijing, China). Amyloglucosidase (CAS No. 9032-08-0, ≥260 U/mL), pancreatin from porcine pancreas (CAS No. 8049-47-6, 8 × USP specifications), invertase (CAS No. 9001-57-4, ≥300 units/mg solids), and pepsin (CAS No. 9001-75-6, ≥250 units/mg solids) were purchased from Sigma-Aldrich Chemical Co., Ltd. (St. Louis, MO, USA). All other chemical reagents were of chromatographic or analytical grade.

### 2.2. Starch Isolation

Starch was isolated according to the modified method of the previous report [21,22]. The materials were milled to pass through a 200-mesh sieve. The powders were steeped in water containing 0.02% NaOH for 12 h. After the precipitation of starch, the supernatant was removed. The starch was washed three times in deionized water. The slurry containing starch was centrifuged in widemouthed cups at 3000 rpm for 5 min. The supernatant and upper non-white layer, which contained the skin and cell wall, were removed. The white layer (starch layer) was resuspended in distilled water and recentrifuged three times. Finally, the starch was washed with ethanol. The starch samples dried overnight at 35 °C and were collected and passed through a 200-mesh sieve.

### 2.3. Chemical Composition

Moisture (method 0832) and ash (method 2302) contents were measured referring to the methods of China Pharmacopoeia. The total starch in each starch sampleswas determined using Solarbio Starch Content Assay Kit (BC0700). Amylose content was determined using Solarbio Amylose Content Assay Kit (BC4260).

### 2.4. Water Solubility and Swelling Power

A total of 200 mg of starch (W) was added to 10 mL of water to make 20% starch emulsion, which was heated at 65 °C, 75 °C, 85 °C, and 95 °C for 1 h, respectively. Then, it was cooled rapidly and centrifuged at 4000× *g* for 15 min. The supernatant was decanted, dried, weighed (DS), and precipitated (SW) after centrifugation, and this was repeated three times [23,24].Water solubility (S) = DS/W × 100%(1)Swelling power (SP) = SW/(W×(100% − S%))(2)

### 2.5. Morphology and Particle Size

Starch microstructure images were observed according to the method described by Liang et al. [25]. The morphology of the starch samples was adhered to conductive adhesive, sprayed with gold, and observed under a 5000 × SEM microscope (Zeiss Sigma 300, Oberkochen, Germany).

The starch granule size distribution was determined using a Mastersizer 2000 laser diffractometer (Mastersizer SALD-2300, Shimadzu, Kyoto, Japan) at room temperature. A total of 0.02 g of starch was suspended in distilled water and fed directly into the instrument’s mixing cell to reach an approximate 2% obscuration level [8].

### 2.6. Molecular Weight

The molecular weight of starch was determined according to Liang et al. [25] with a minor modification. The molecular weight distribution of debranched starch was analyzed by a gel chromatography–differential-multi-angle laser light scattering system. The differential detector is an Optilab T-rEX laser light scattering detector (Wyatt Technology, Santa Barbara, CA, USA). According to the properties of the compound, the gel exclusion chromatographic columns of Ohpak SB-805 HQ (300 × 8 mm, Wyatt Technology, Santa Barbara, CA, USA), Ohpak SB-804 HQ (300 × 8 mm, Wyatt Technology, Santa Barbara, CA, USA), Ohpak SB-803 HQ (300 × 8 mm, Wyatt Technology, Santa Barbara, CA, USA), and column temperature of 60 °C were used for DAWN HELEOS II (Wyatt Technology, Santa Barbara, CA, USA). The injection volume was 200 μL, and the mobile phase of DMSO containing 0.5% LiBr was used. The flow rate was 0.3 mL/min.

### 2.7. Crystal Structure

Using the procedure described by Zhao et al. [26], with slight modifications, the crystalline patterns of the starches were analyzed with an X-ray diffractometer (SHIMADZU, Kyoto, Japan) with Cu-Kα radiation at 40 kV and 30 mA. The scanning angle range was from 5° to 45°, and the scanning speed was 2°/min. The relative crystallinity was calculated by Jade 6.0 (Materials Data Inc., Livermore, CA, USA).

### 2.8. Fourier Transform Infrared (FT-IR) Spectroscopy

Fourier transform infrared spectra were determined using an FTIR spectrophotometer (Thermo Nicolet iS5, Waltham, MA, USA). The FT-IR spectra of the samples were acquired by the potassium bromide compression method. For this purpose, an appropriate amount of potassium bromide was mixed with the dried sample powder (1 mg), and the mixture was subsequently milled until free of granularity. The samples were then pressed into tablets for processing. The wavelength range was between 4000 and 400 cm^−1^, with 32 scans and a resolution of 4 cm^−1^ [27].

### 2.9. Thermal Stability Analysis (TGA)

The thermal stability determination of starches was conducted with a simultaneous NETZSCH TG 209F3 thermal analyzer (NETZSCH, Selb, Germany). The samples (3–5 mg) were heated from 30 °C to 700 °C at a heating rate of 10 °C/min.

### 2.10. Determination of Pasting Properties

The pasting properties of the samples were analyzed using the Rapid Visco-Analyzer (Model RVA Super 4, Perten, Bonn, Swedish), according to Kumar et al. [28]. Appropriate samples were weighed into the RVA canister. The sample suspension was mixed by hand using a stirring paddle for 30–40 s to avoid the formation of lumps before the RVA run. A prespecified cycle of heating and cooling was followed. The sample suspension was held at 50 °C for 1 min, heated to 95 °C within a timeframe of 4 min 45 s, and then maintained for 2 min 30 s at 95 °C. The sample was then cooled to 50 °C within 3 min 48 s and held at 50 °C for an additional 2 min. The RVA speed of the internal stirring arm was kept constant at 160 PM throughout the run. It should be noted that the value of the RVA is given by the Rapid Visco Unit (RVU).

### 2.11. In Vitro Digestion of Starch

The digestion properties of PM and PMP starches were carried out based on a previously described method [29,30]. The samples (200 mg) were fully mixed with deionized water (2 mL) using a magnetic stirrer. The mixtures were then cooked for 20 min in a boiling water bath. Afterwards, the gelatinized samples and pepsin/hydrochloric acid solution (4 mL, 5 mg/mL) were incubated at 37 °C with shaking for 30 min. Thereafter, six glass beads (3–4 mm diameter) and sodium acetate buffer (2 mL, 0.5 M, pH 5.2) were added to reaction mixtures, and the reaction was continued for 30 min. Starch digestion was initiated with the addition of 2 mL enzyme mixtures (pancreatin extract, amyloglucosidase, and invertase). After enzymatic digestion at specific time points (0, 20, and 120 min), aliquots (50 μL) of the hydrolysate were withdrawn and mixed with 950 μL of 75% (*v*/*v*) ethanol solution to stop the digestive reaction. After centrifuging at 10,000 PM for 5 min, the supernatant was used to measure the hydrolyzed glucose content by the glucose assay kit (GOPOD format). The starch component content of test samples was calculated according to a method described by Li et al. [31].

### 2.12. Animal Experimental Design

The animals used in this experiment were specific pathogen-free (SPF) ICR mice (male, 7–8 weeks old, 18–20 g in body weight) purchased from SPF (Beijing, China) Biotechnology Co., Ltd. (animal license number: SCXK (Beijing, China) 2023–0011). The mice were kept in the animal room of the Beijing University of Traditional Chinese Medicine (BUCM). Before the start of the formal experiment, the mice were adaptively fed in advance for one week. All animal experiments were approved by the Institutional Animal Care and Use Committee (IACUC) of BUCM (protocol number: BUCM- 2,023,101,701–4120, date: 20 November 2023). And the methods were carried out in accordance with the approved guidelines.

A total of 65 ICR mice were randomly divided into seven groups: control (n = 5), PM starch (0.15 g/mL) (PM1, PM2, and PM3, n = 10), and PMP starch (0.15 g/mL) (PMP1, PMP2, and PMP3, n = 10) groups. The mice in the control group were given equal doses of normal saline (1 mL/kg). The sample solution was administered by gavage on five consecutive days. The mice were euthanized 2 h after intragastric administration on the fifth day, and feces were collected.

### 2.13. Gut Microbiota Analysis

Microbial community genomic DNA was extracted from randomly selected six fecal samples from each group using an E.Z.N.A.^®^ DNA Kit (Omega Bio-Tek, Norcross, GA, USA). The DNA extract was electrophoresed on an agarose gel, and the DNA concentration was determined using a NanoDrop 2000 UV–vis spectrophotometer (Thermo Scientific, Waltham, MA, USA). DNA was extracted and collected from the fermented slurry. The quantity and quality of the extracted DNA were measured using an ultraviolet spectrophotometer and agarose gel electrophoresis, respectively. Polymerase chain reaction (PCR) was used to amplify the V3–V4 region of the 16S rRNA gene for each sample. The forward primer 338F (5′-barcode+ACTCCTACGGGAGGCAGCA-3′) and the reverse primer 806R (5′-GGACTACCAGGGTATCTAAT-3′) were employed during PCR amplification. The PicoGreen dsDNA Assay Kit was used to quantify amplicons. After individual quantification, amplicons were pooled in equal amounts, and sequencing was performed on the Illumina MiSeq platform (Shanghai Personal Biotechnology Co., Ltd., Shanghai, China). All of the results, including the determination of α- and β-diversity, were subjected to species screening based on operational taxonomic units [32].

### 2.14. Statistical Analysis

All experiments were performed in triplicate, and data were presented as mean ± standard deviation (SD). The Ducan test method in the analysis of variance (ANOVA) was applied for the significance analysis. *p* < 0.05 denoted the significance level of the test. Pearson correlation coefficients and principal component analysis (PCA) were performed with the SPSS 20.0 software. Excel 2010 and Origin 8.0 were used to process and analyze data.

## 3. Results

### 3.1. Comparison of the Chemical Composition of Different Starches

The functional properties of starch are significantly influenced by its chemical composition, which is an important intrinsic factor. The measurements of moisture content, ash content, starch content, AM content, solubility, and swelling power of PM and PMP starches are summarized in Table 1. The values of moisture content of PM starch ranged between 7.81% and 9.32%, and the PMP starch ranged between 8.44% and 9.79%. The values of ash content of PM starch ranged between 4.22% and 4.93%, and the PMP starch ranged between 2.54% and 2.97%. The total starch content of PM ranged between 127.97% and 197.83%, and PMP ranged between 113.51% and 197.27%. The amylose content of PM ranged from 82.30% to 88.22%, and PMP ranged from 72.81% to 79.16%. The moisture content, ash content, total starch content, and amylose content of PM and PMP starches had less obvious change, indicating that processing of Polygonum multiflorum did not affect the basic properties of these starches.

The swelling power and solubility of PM and PMP starches are presented in Table 1. Among PM starch, swelling powers ranged from 12.71 to 15.86, and solubilities ranged from 2.79% to 2.89%. For PMP starch, swelling powers ranged from 6.04 to 8.62, and solubilities ranged from 4.04% to 4.74%. It has been reported that amylose content could affect starch swelling power and water solubility [33,34]. Amylose, by interaction with the amylopectin backbone, presents a major retarding influence on the swelling power of the starch granules [35].

### 3.2. Morphological Analysis

SEM was employed to analyze the morphological characteristics of PM starch and PMP starch. The results revealed that there were highly significant differences in the appearance and structure of the starch granules between the two types (Figure 1). The PM starch particles are polygonal, irregular, and spherical with smooth surfaces, as mentioned in previous studies [21,22]. Nevertheless, the PMP starch swelled and shrank to form lumps with cracks and holes in surfaces. The soaking and cooking processes likely played a role in this. These processes induced the gelatinization and expansion of PM starches. As a result of such changes, an amorphous appearance was created. With the increase of the times of steaming and drying, the starch broke up, forming clumps with different holes in the surface. During the processing, notable morphological alterations occurred. The PM starch underwent a process where its original form was entirely disintegrated, and the natural crystalline structure was thoroughly shattered. Consequently, the resultant starch took on irregular shapes. In terms of morphological characteristics, the PM starch bore a resemblance to pregelatinized starch. [36]. The average particle size of PM and PMP starches are shown in Table 2, which demonstrates that the differences in PM and PMP starch particle size and diameter distribution were very significant. D (0.25), d (0.50), and d (0.75) refer to 25%, 50%, and 75% of the particle size in the measured size values, respectively. The mean particle size of PM ranged between 13.876 and 17.105, but the PMP ranged between 41.350 and 54.412. The latter is far greater than the former. These results also reflect that PM starch cannot be dispersed into uniform granules after processing. The variation in starch particle size may be related to the process during starch preparation, which was confirmed by SEM.

### 3.3. Molecular Weight and Pasting Properties

Table 3 presents the Mw (weight-average molecular weight), Mn (number-average molecular weight), and Mw/Mn (D, the degree of dispersion of the molecular weight distribution) values for PM and PMP starches. In the context of Mw/Mn (D), a D value of 1 indicates a polymer with a uniform molecular weight. As the D value increases, it implies a broader molecular weight distribution and higher polydispersity. It was observed that the D value of PM starch exceeded that of PMP starch. This difference can be attributed to the processing steps of steaming and drying. During gelatinization, these processing conditions cause the dissociation of the double helix in amylopectin and the stripping of amylose. Ultimately, this series of events leads to an alteration in the molecular weight of the starch, as presented in Table 3.

The pasting parameters of PM and PMP starch samples are given in Table 4. The pasting temperature of PM starch was between 72.35 °C and 74.80 °C. Due to the fact that PMP is obtained from the process of nine steaming and nine drying of PM, starch has already undergone gelatinization during the steaming and drying process, and no gelatinization temperature was detected. After processing, it was found that the values of peak viscosity (PV), breakdown (BD), final viscosity (FV), and setback (SB) decreased significantly in PMP starch, whereas the values of peak time increased compared with PM starch. This could be mainly due to a reduction in the swelling power and solubility of PMP starch by steaming and drying compared with PM starch, respectively (Table 1). The lower swelling power resulted in a decrease in PV and BD values, whereas the lower solubility led to a decrease in FV and SB values of PMP starch. BD measures how easily starch granules are affected by thermal shear forces, and SB reflects the likelihood of starch paste reverting to its original, less-soluble state (retrogradation). When starch is being heated, if the starch granules do not swell much, they are less likely to be broken down by thermal shear. As a result, the BD value will be lower. During the cooling process (continuously), when less starch (particularly amylose) is dissolved and released, there is less starch available to reassociate. This leads to a lower SB value [28,37].

### 3.4. X-Ray Diffraction

In the natural environment, starch is present in the form of semi-crystalline granules. The crystalline structure of starch depends on multiple aspects, such as its origin, chemical composition, physical morphology, molecular features, and the way its molecules are arranged. Moreover, this crystalline structure serves as a crucial determinant of the functional attributes exhibited by starch [38]. XRD patterns of the PM and PMP starches are presented in Figure 2A. According to the X-ray diffraction pattern, starch can be divided into A, B, C, and V types [26,39,40]. Compared with the standard diffraction patterns [41,42], PM starch displayed a typical B-type pattern, with strong reflection at 2θ 15°, 17°, 18°, and 23°. This is consistent with a previous study [21]. Notably, the XRD patterns of PMP starch underwent substantial alterations. PMP starch displayed diffraction peaks at angles of 10.0°, 20.5°, and 26.5°. These peaks are indicative of the V-type structure typical of amorphous crystals. Conversely, the diffraction peaks that were initially present at 15°, 17°, 18°, and 23° vanished. This disappearance can be ascribed to the steam-drying process employed, which disrupted the internal structure of the starch granules. This change provides evidence that the starch has undergone pasting. Additionally, the emergence of the V-type crystalline structure aligns with one of the defining features of pregelatinized starch, as previously reported in a reference [43].

The relative crystallinity (RC) of starches is presented in Table 3. The results showed that the PMP starch increased compared to that of PM starch. This phenomenon can be explained by the preferential hydrolysis of the starch’s amorphous region. As the amorphous part is hydrolyzed, the relative proportion of the crystalline region within the starch increases. During the steaming procedure, the amylose and amylopectin chains degrade, giving rise to short and disordered chains. These chains then undergo rearrangement, ultimately resulting in the formation of weakly crystalline structures, as noted in previous studies [44,45].

### 3.5. FT-IR Analysis

FT-IR spectroscopy was carried out on PM starch and PMP starch to explore their molecular interactions. As depicted in Figure 2B for PM starch, the peaks within the 3600–3000 cm⁻^1^ range were due to the O-H stretching vibrations inherent to starch. The band at 1635 cm⁻^1^ was linked to the water absorbed in the starch’s amorphous region. Moreover, the absorbance values at 1047 cm⁻^1^ and 1022 cm⁻^1^ corresponded to the crystalline and amorphous structures of starch, respectively, as reported in previous studies [46,47,48]. Turning to the PMP starch, its FT-IR spectrum in Figure 2B exhibited distinct signals. Alongside the characteristic O-H vibrations in the 3600–3000 cm⁻^1^ area, the signals at 1655 cm⁻^1^ and 1556 cm⁻^1^ bands originated from the C=O and C=C stretching vibrations [49,50]. These indicated the existence of complexes within the PMP starch [27].

Table 3 presents the absorbance ratio of 1047/1022 cm^−1^ for all starches. The absorption values at 1047 cm^−1^ and 1022 cm^−1^, respectively, reflect changes in the crystalline and amorphous regions of starches. This 1047/1022 cm^−1^ absorbance ratio is capable of disclosing the short-range molecular ordering close to the surface of starch granules [51]. When compared to PM starch, PMP starch exhibited a slightly elevated 1047/1022 cm^−1^ ratio. This difference can be accounted for by the molecular arrangement within the starch granules. It implies that during the heating process, there is a more efficient organization of the double-helical structure in the outer part of the starch granules of PMP starch, resulting in a greater degree of short-range molecular order. A comparable tendency has been noted in other studies on heat-moisture-treated starches from different sources [28]. The crystallinity degree results obtained from FTIR analysis are consistent with those of relative crystallinity derived from XRD analysis.

### 3.6. Thermal Stability Analysis

Figure 3 presents the TGA and derivative thermogravimetric (DTG) curves of PM starch and PMP starch. Here, the DTG curve indicates the rate of weight loss. As can be seen from Figure 3, all starch samples underwent three weight-loss stages. In the first stage, occurring within a temperature range of 30 to 120 °C, the weight loss was attributed to the evaporation of bound water. The second prominent weight-loss stage, taking place between 200 and 400 °C, was a result of starch decomposition, as previously reported [52]. The three-stage mass loss occurring from 400 °C to 700 °C was due to the thermal degradation of carbonaceous residue remains [53]. The results showed that the Tmax of PM starch was slightly higher than PMP starch. As described in Figure 4, the temperature value of PM starch occurred at 324.9 °C, 320.9 °C, and 319.0 °C, respectively. The temperature value of PMP starch occurred at 317.6 °C, 319.5 °C, and 308.0 °C, respectively. This behavior suggests that the thermal stability of PM starch decreased in the process of steaming.

### 3.7. In Vitro Digestion Property

The starches digested within 20 min, from 20 to 120 min, and remained after 120 min were commonly defined as RDS, SDS, and RS, respectively [31]. The variation in the digestibility outcomes among the starch samples can be ascribed to the combined influence of multiple factors. These include granule dimensions, amylose concentration, the distribution pattern of amylopectin unit chains, the organization within the amorphous and crystalline regions of starch, and the degree of crystallinity, as well as the presence of pores and channels in the granules. Furthermore, the size of starch granules plays a role in determining digestibility. This is because the surface area and volume of the granules impact the interaction between the starch substrate and the enzyme. The digestion properties of PM starch and PMP starch are displayed in Table 5. The RDS, SDS, and RS of PM starch ranged from 36.30% to 38.29%, from 4.53% to 16.04%, and from 45.66% to 58.35%, respectively. While the RDS, SDS, and RS of PMP starch ranged from 10.60% to 13.10%, from 21.75% to 26.69%, and from 60.21% to 67.19%, respectively. The SDS and RS content of PMP starch increased significantly (*p* ≤ 0.05), while the RDS content decreased significantly (*p* ≤ 0.05) compared with PM starch. This result could be attributed to the reduction of the amorphous chains of the starch granules. The result is consistent with the previous XRD results. The steam treatment acted mainly on the amorphous zones of the starch granules and consequently increased the proportion of crystalline regions, thus causing an increase in the SDS and RS content [54,55].

### 3.8. Microbial Community Distribution

16S rRNA gene sequencing was performed on fecal samples to further explore the effects of PM and PMP starches on the gut microbiota composition. The rarefaction curve is generated by randomly selecting a certain number of sequences from the sample and then calculating the corresponding Alpha diversity index for these sequences. The number of sequences sampled is plotted on the *x*-axis, and the Alpha diversity index value on the *y*-axis forms the curve. The curve’s flattening indicates whether the sequencing depth is sufficient. As shown in Figure 4A,B, both indices’ rarefaction curves have plateaued, suggesting that the sequencing depth in this experiment is adequate. Additional data would only yield a minimal number of new species. This indicates that the sequencing results are reliable and capture the majority of the microbial community information present in the sample.

Diversity analysis was conducted on samples from three groups (blank control group, PM starch group, and PMP starch group) using 16S rRNA sequencing. A total of 3,479,726 sequences were detected, with an average sequence length of 421 bp. The non-redundant sequences were clustered at 97% similarity, resulting in 1641 OTUs, covering 11 phyla, 16 classes, 47 orders, 75 families, 158 genera, and 305 species. The Venn diagram was used to compare the number of shared and unique species across multiple groups or samples, providing a clear visualization of the species composition similarity and overlap between different samples. As shown in Figure 4D, the blank control group had 37 unique OTUs, the PM starch group had 274 unique OTUs, and the PMP group had 226 unique OTUs. The number of unique OTUs in the PM group was higher than that in the PMP group and the blank control group, indicating a greater diversity of bacterial species in the PM starch group.

At the genus classification level, the intestinal microbiota of mice in all groups included a total of 158 genera. The species with the high abundance at the genus level were *norank_f__Muribaculaceae*, *Lactobacillus*, *Bacteroides*, *Prevotellaceae_UCG-001*, *Lachnospiraceae_NK4A136_group*, *Eubacterium_fissicatena_group*, *Dubosiella*, *Alloprevotella*, *unclassified_f__Lachnospiraceae*, and *Rikenella*. As shown in the microbial community distribution map and heatmap (Figure 4E,F), there are distinct differences in the genus composition of the intestinal microbiota among the groups of mice. In the heatmap, the redder the color block, the higher the relative abundance of the genus, while the bluer the block, the lower the relative abundance. The relative abundance of *norank_f__Muribaculaceae* in the intestinal microbiota of mice from the blank control group, PM starch group, and PMP starch group were 33.08%, 40.35%, and 34.02%, respectively. The relative abundance percentages of *Lactobacillus* in the three groups were 13.61%, 12.42%, and 16.26%, respectively. Overall, oral administration of PM and PMP starches to mice led to some differences in the composition of their intestinal microbiota, increasing the evenness of bacterial genera and, to some extent, enhancing the diversity of the gut microbiota.

As shown in Figure 5A, the species–sample relationship diagram indicates that the PM starch group has a significant proportion of *Proteobacteria*, *Campilobacterota*, *Deferribacterota*, *Verrucomicrobiota,* and *Bacteroidota*. Meanwhile, Firmicutes hold an important proportion in the PMP starch group. Significant differences were observed among the three groups of samples in *Arthromitus*, *Blautia*, *norank_f_Desulfovibrionaceae*, *Colidextribacter*, *Muribaculum*, *Eubacterium*, *Lachnospiraceae*, *Bifidobacterium*, *Erysipelotrichaceae*, and *Desulfovibrio* (Figure 5B). Especially *Blautia*, which exhibits significant differences, has the ability to undergo biotransformation and plays a role in regulating host health and alleviating metabolic syndrome [56]. To identify specific microbial taxa at the genus and family levels associated with the gut microbiota interventions of raw and processed *Polygonum multiflorum* starch, we performed LEfSe analysis of the microbiota communities. As shown in Figure 6A,B, we identified 26 significantly enriched taxa in two groups. The PM group was predominantly characterized by the genera *Proteobacteria*, *Gammaproteobacteria*, *Burkholderiales*, *Sutterellaceae*, and *Parasutterella*. In contrast, the PMP group was characterized by *Clostridia*, *Lachnospiraceae*, and *Lachnospirales*. *Clostridium* plays an important role in the prevention, diagnosis, and treatment of diseases. It serves as a potential biomarker for certain diseases, helps maintain the balance of the gut microbiota, enhances the function of the intestinal mucosal barrier, and acts as a target for drug development. *Lachnospiraceae* participates in the metabolism of various carbohydrates, with fermentation-producing acetate and butyrate as the primary energy sources for the host. *Lachnospirales* has been shown to prevent and treat both intestinal and extra-intestinal diseases. Therefore, these bacteria may become crucial microbiota for the prevention and treatment of gut-related diseases, which helps explain the important role of PMP starch in disease treatment.

## 4. Discussion

The structural and functional divergences observed between raw (PM) and processed *Polygonum multiflorum* (PMP) starches underscore the profound impact of traditional processing on starch physicochemical properties and bioactivity. While the amylose content showed no significant difference between PM and PMP starches—a finding consistent with some studies on heat-modified starches—the stark contrast in swelling power and solubility aligns with prior hypotheses that thermal processing disrupts granular architecture and chain interactions. The marked reduction in swelling power and increased solubility can be attributed to gelatinization-induced chain fragmentation and granule collapse, as proposed in earlier models of starch retrogradation and molecular reorganization. This phenomenon mirrors observations in other processed tuber and cereal starches, where thermal treatment disrupts hydrogen bonding and amylopectin double helices, reducing water-holding capacity while enhancing the solubilization of shorter amylose fragments. The SEM and XRD data provide compelling evidence for structural disintegration in PMP starch, transitioning from intact, smooth granules to porous aggregates—a pattern previously linked to hydrothermal processing in starches like potato and cassava. The loss of crystalline order (evidenced by diminished XRD peaks) and the FTIR-detected shift toward short-range ordered structures suggest that processing not only destroys long-range crystallinity but also promotes localized reordering, a paradox observed in gelatinized starches where amorphous regions dominate yet retain residual molecular alignment. These structural changes correlate with reduced thermal stability in PMP starch, a trend reported in modified starches with disrupted granular integrity, where weakened molecular packing lowers resistance to thermal degradation.

The significant increase in resistant starch (RS) content in PMP starch aligns with the “structural barrier” hypothesis. As shown by scanning electron microscopy, the aggregated lumps and porous surfaces of PMP starch limit enzyme access. This restricts digestion, thereby increasing RS production. Similar to studies on retrograded or physically modified starches, the enhanced structural recalcitrance in PMP starch boosts its potential for colonic fermentation.

The regulation of gut microbiota observed in mice, especially the enrichment of *Clostridia*, *Lachnospiraceae*, and *Lachnospirales*, corresponds with findings related to RS and butyrate-producing taxa. Butyrate-producing taxa play a crucial role in maintaining intestinal barrier integrity and modulating anti-inflammatory responses. RS serves as a favorable substrate for these taxa. In the large intestine, through a series of metabolic pathways, bacteria break down RS. For instance, some species of *Clostridia* and *Lachnospiraceae* possess specific enzymes that can ferment RS. They convert RS into short-chain fatty acids, with butyrate being a key product. This process not only provides energy for the bacteria but also benefits the host’s intestinal health. This indicates that the bioactivity of PMP starch may stem from its role as a microbial substrate, and processing likely enhances this bioactivity. From a broader perspective, these findings confirm that traditional processing methods represent a form of “natural engineering” that can enhance the functional properties of herbal starches. The structure–function relationships elucidated in this paper are not limited to Polygonum multiflorum but are also applicable to other medicinal plants. In these plants, starch modification may underlie the efficacy of processing.

## 5. Conclusions

This study systematically characterized and compared the structure and physicochemical properties of starch before and after processing Polygonum multiflorum. Significant differences were observed between the starches from PM and PMP, primarily due to starch gelatinization during processing, which led to changes in the starch properties. Additionally, thermal processing increased the RS content by 7% to 22% and enriched beneficial gut microbes, such as *Clostridia*, *Lachnospiraceae*, and *Lachnospirales* species, which play a role in promoting gut health. The starch processing–structure–function relationship revealed in this study offers a multi-level optimization pathway for developing functional foods from traditional Chinese medicines. By precisely regulating thermal processing parameters, it is possible to achieve the targeted design of starch solubility and swelling power, providing a foundational material engineering approach for functional food production. Moreover, leveraging the increase in resistant starch content induced by processing, we can develop a carbohydrate-based matrix with controlled-release properties. Finally, with the specific modulation of intestinal flora by PMP starch, we can integrate it with probiotics to develop symbiotic food systems with gut–brain axis regulation functions. To advance the application of these findings, future research should focus on addressing structural-function stability issues in traditional drug–food matrices and optimizing consumer acceptability through sensory analysis techniques, ultimately achieving the organic integration of traditional medicinal knowledge with modern food science.

## Figures and Tables

**Figure 1 foods-14-01578-f001:**
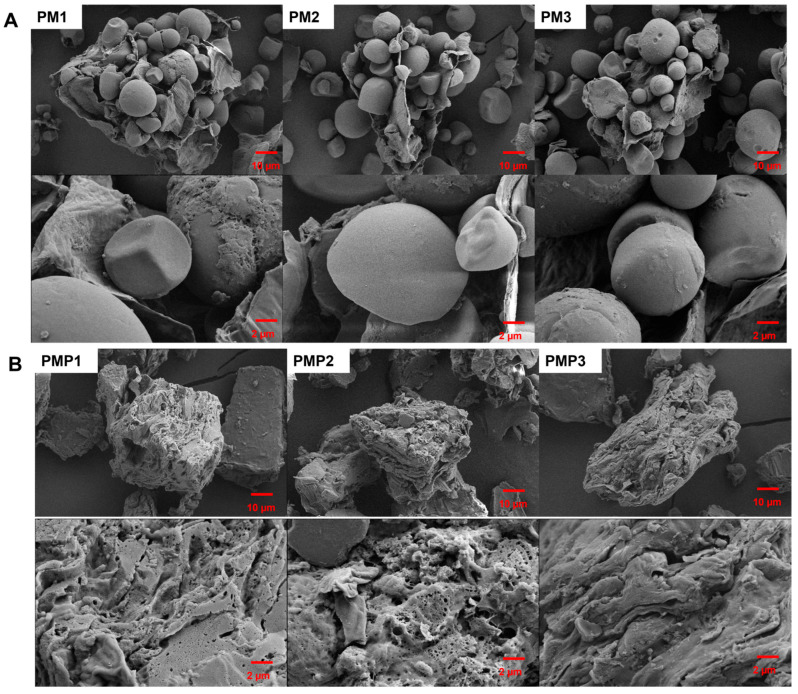
SEM observation of starch isolated from PM (**A**) and PMP (**B**).

**Figure 2 foods-14-01578-f002:**
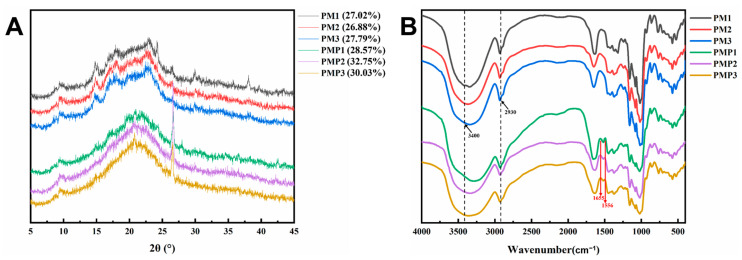
(**A**) XRD spectra of starch isolated from PM and PMP; (**B**) FT-IR transmittance spectra of PM starch and PMP starch.

**Figure 3 foods-14-01578-f003:**
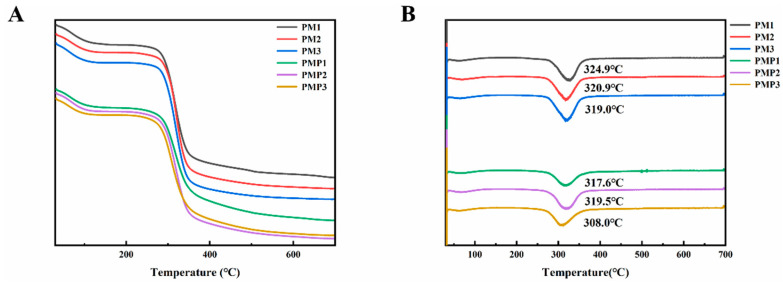
TGA (**A**) and DTG (**B**) curves of PM starch and PMP starch.

**Figure 4 foods-14-01578-f004:**
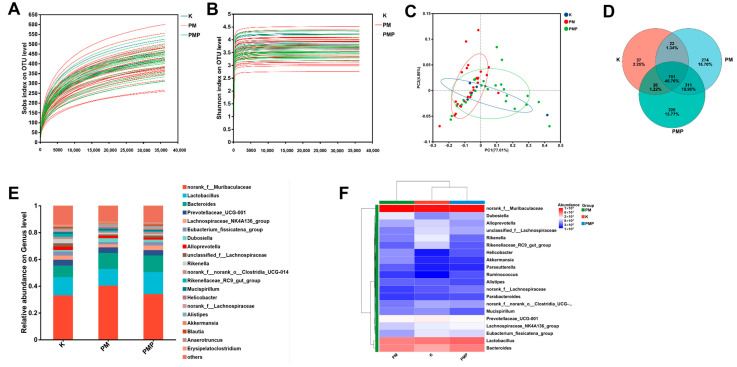
Diversity analysis of flora; (**A**,**B**) rarefaction curve at the OTU level; (**C**) principal coordinates analysis (PCoA) based on phylum level; (**D**) Venn diagram; (**E**) genus level community composition of flora; (**F**) heatmap of community composition of flora.

**Figure 5 foods-14-01578-f005:**
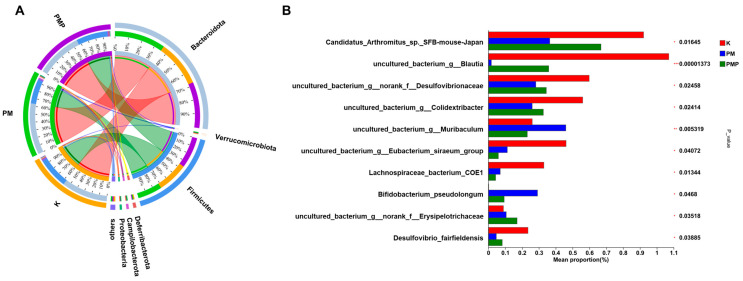
(**A**) Diagram of the relationship between Circos samples and species; (**B**) species level multi-group comparison chart. (* *p* < 0.05, ** *p* < 0.01 and *** *p* < 0.001)

**Figure 6 foods-14-01578-f006:**
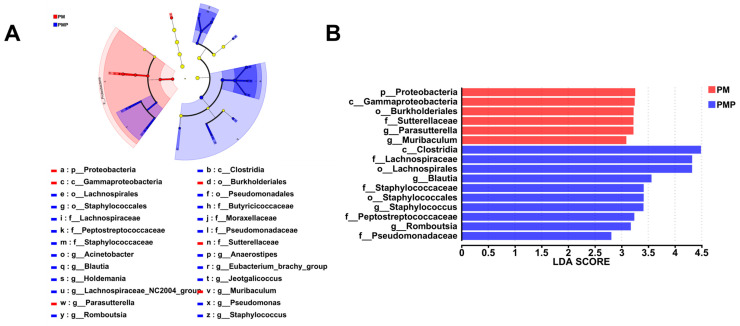
Identification of most characteristic taxa among K, PM, and PMP groups by linear discriminant analysis (LDA) effect size (LEfSe). (**A**) Cladogram visualizing the output of the LEfSe analysis. (**B**) Most significant difference of gut microbial taxa among groups after LDA.

**Table 1 foods-14-01578-t001:** Moisture contents, ash contents, starch contents, AM contents, solubility, and swelling power of starch isolated from PM and PMP. (n = 3).

Sample	Moisture Content (%)	Ash Content (%)	Total Starch Content (mg/g)	Amylose Content (mg/g)	Swelling Power	Solubility (%)
PM1	9.32 ± 0.03	4.93 ± 0.01	197.83 ± 6.98	87.70 ± 6.12	12.71 ± 1.68	2.79 ± 0.11
PM2	7.81 ± 0.01	4.22 ± 0.00	127.97 ± 5.97	88.22 ± 5.32	15.86 ± 0.84	2.43 ± 0.08
PM3	8.61 ± 0.01	4.68 ± 0.00	147.76 ± 2.04	82.30 ± 2.16	14.49 ± 0.96	2.89 ± 0.02
PMP1	9.54 ± 0.01	2.68 ± 0.02	197.27 ± 6.47	72.81 ± 2.87	6.04 ± 0.05	4.74 ± 0.14
PMP2	9.79 ± 0.01	2.54 ± 0.00	140.14 ± 5.62	74.99 ± 3.00	7.89 ± 0.09	4.87 ± 0.17
PMP3	8.44 ± 0.01	2.97 ± 0.00	113.51 ± 1.87	79.16 ± 1.40	8.62 ± 0.29	4.07 ± 0.12

**Table 2 foods-14-01578-t002:** Particle size of starches from PM and PMP. (n = 3).

Sample	d (0.25) (μm)	d (0.50) (μm)	d (0.75) (μm)	Median Particle Size	Mean Particle Size	Modal Particle Size
PM1	10.95 ± 0.25	18.86 ± 0.09	32.73 ± 0.32	18.86 ± 0.25	17.105 ± 0.42	17.14 ± 0.12
PM2	10.05 ± 0.08	15.57 ± 0.12	24.08 ± 0.36	15.57 ± 0.26	13.876 ± 0.40	17.14 ± 0.16
PM3	10.57 ± 0.09	18.48 ± 0.23	34.36 ± 0.35	18.48 ± 0.34	16.960 ± 0.44	13.58 ± 0.09
PMP1	39.71 ± 0.14	56.38 ± 0.26	77.45 ± 0.32	56.38 ± 0.45	54.412 ± 0.25	54.92 ± 0.35
PMP2	30.11 ± 0.15	46.39 ± 0.42	67.91 ± 0.33	46.39 ± 0.56	41.350 ± 0.35	54.92 ± 0.32
PMP3	29.58 ± 0.12	45.84 ± 0.45	67.51 ± 0.35	45.84 ± 0.34	41.374 ± 0.34	54.92 ± 0.36

**Table 3 foods-14-01578-t003:** FT-IR, XRD characteristic, and Mw of PM starch and PMP starch.

Sample	Mw (kDa)	Mn (kDa)	Mw/Mn	Relative Crystallinity (%)	Ratio at 1047/1022 cm^−1^
PM1	24,236.474	7645.310	3.170	27.20	0.90
PM2	44,421.388	12,225.316	3.634	26.88	0.90
PM3	25,292.823	7300.397	3.465	27.79	0.93
PMP1	2158.057	1488.433	1.450	28.57	0.95
PMP2	976.872	551.430	1.772	32.75	0.95
PMP3	279.093	176.865	1.578	30.03	0.94

**Table 4 foods-14-01578-t004:** Pasting properties of PM starch and PMP starch. (n = 3).

Sample	Peak Viscosity (PV)	Trough Viscosity (Pa⋅s)	Breakdown (BD)	Final Viscosity (FV)	Setback (SB)	Peak Time	Pasting Temp
PM1	634 ± 0.82	409 ± 1.63	225 ± 2.45	586 ± 3.27	177 ± 0.47	4.00 ± 0.05	74.05 ± 0.0
PM2	1139 ± 2.05	667 ± 4.08	472 ± 3.27	1050 ± 4.08	383 ± 2.45	3.80 ± 0.05	72.35 ± 0.41
PM3	880 ± 4.90	521 ± 3.27	359 ± 3.27	794 ± 4.08	273 ± 3.27	4.00 ± 0.09	74.80 ± 0.57
PMP1	16 ± 0.8	15 ± 0.82	1 ± 0.00	19 ± 0.47	4 ± 0.47	4.07 ± 0.06	-
PMP2	47 ± 2.45	44 ± 0.47	3 ± 0.47	78 ± 0.47	34 ± 0.47	6.73 ± 0.05	-
PMP3	19 ± 0.82	18 ± 0.47	1 ± 0.00	29 ± 0.82	11 ± 0.47	6.07 ± 0.06	-

**Table 5 foods-14-01578-t005:** In vitro digestibility of PM and PMP starches (n = 3).

Sample	RDS (%)	SDS (%)	RS (%)
PM1	37.11 ± 0.47	4.53 ± 0.11	58.35 ± 1.27
PM2	36.30 ± 0.33	13.10 ± 0.16	50.60 ± 1.27
PM3	38.29 ± 0.30	16.04 ± 0.29	45.66 ± 1.33
PMP1	13.10 ± 0.17	21.75 ± 0.63	65.15 ± 0.95
PMP2	13.10 ± 0.15	26.69 ± 0.99	60.21 ± 1.05
PMP3	10.60 ± 0.41	22.21 ± 1.30	67.19 ± 0.74

RDS: slowly digestible starch; SDS: rapidly digestible starch; RS: resistant starch.

## Data Availability

The original contributions presented in the study are included in the article, and further inquiries can be directed to the corresponding authors.

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
