# Peer review of "Comparison of the Structure, Physicochemical Properties, and Impact on Intestinal Flora of Processed and Unprocessed *Polygonum multiflorum* Starch"

_foods, 2025, doi:10.3390/foods14091578_

Round 1
Reviewer 1 Report
Comments and Suggestions for Authors
The manuscript is very interesting and well written. The part related to starch chemistry is well presented and discussed. However, the part related to microbiota modulation needs some adjustments.
Please, find some details below.
Lines 33-34 "Amylopectin, on the other hand, 33 consists of numerous smaller amylose molecules connected by α-1,6 glucoside bonds to 34 form branched structures, with a molecular weight ranging from 1×10⁵ to 1×10⁶ Da." Please, include a citation.
Lines 56-57 "However, it can be fermented or partially me-56 tabolized by bacteria in the large intestine to produce short-chain fatty acids, lactic acid, 57 and a small amount of gas."Please, include a citation.
Lines 68, 69, 75, 78 and all over the manuscript "Polygonum multiflorum" - Italic
Line 92 "in vitro" - Italic
Lines 178-179 About digestion, why did you not choose INFOGEST method?
Line 198 "All animal experi ments" Please, correct the word experiments.
Line 213. "Gene sequencing and data analysis were conducted following a previously described method (Huang et al., 2022)". Please, describe it breafly.
Table 5 Please, include in the title the meaning for RDS, SDS and RS.
Line 410 "Lactobacillus". Italic
Lines 419-420 and all over the manuscript "Arthromitus, Blautia, norank_f_Desulfovibrionaceae, Colidextribacter, 419 Muribaculum, Eubacterium, Lachnospiraceae, Bifidobacterium, Erysipelotrichaceae, and Desul-420 fovibrio (Figure 4G)." Family taxonimic level (ex. Lachnospiraceae, Erysipelotrichaceae) is not written in italic. Please correct change for italics to not italic.
Line 429 "Relevant studies have confirmed that Clostridia plays a role in disease prevention, diagnosis, and treatment." This statmente is confuse. Can you better explain how Clostria can be relevant to diagnosis and treament?
Figure 4 - On letter D, there is not only Genus level expressed, as stated in axis y. Please, correct it.
Figure 4 - Letter C, E, F and G
PM is Polygonum multiflorum Thunb.
PMP is processed Polygonum multiflorum
What is K?
Lines 483-484 "Clostridia, Lachnospiraceae, and Lachnospirales species, which play a role in promoting gut health." An increase in Clostridia is not always a good outcome. Many pathogens belong to the genus Clostridium, such as Clostridium butulinum, Clostridium perfrigens and Clostridium difficile.
Comments on the Quality of English Language
Good quality
Author Response
Please see the attachment.
|
Response to Reviewer 1 Comments |
||
|
1. Summary |
|
|
|
Thank you very much for taking time out of your busy schedule to review this manuscript. We have made the modifications as per your suggestions, and all the modified parts have been highlighted in yellow. Thank you again for your suggestion. |
||
|
2. Point-by-point response to Comments and Suggestions for Authors |
||
|
Comments 1: Lines 33-34 "Amylopectin, on the other hand, 33 consists of numerous smaller amylose molecules connected by α-1,6 glucoside bonds to 34 form branched structures, with a molecular weight ranging from 1×10⁵ to 1×10⁶ Da." Please, include a citation. |
||
|
Response 1: Thank you for pointing this out. We agree with this comment. Therefore, we have added reference [2] in line 35. |
||
|
Comments 2: Lines 56-57 "However, it can be fermented or partially me-56 tabolized by bacteria in the large intestine to produce short-chain fatty acids, lactic acid, 57 and a small amount of gas." Please, include a citation. Response 2: Thank you for pointing this out. We agree with this comment. Therefore, we have added reference [9] in line 58. |
||
|
Comments 3: Lines 68, 69, 75, 78 and all over the manuscript "Polygonum multiflorum" – Italic. |
||
|
Response 3: Thank you for pointing this out. We agree with this comment. Therefore, We have changed the Polygonum multiflorum format to italics. (line 70, 71, 80, 95) Comments 4: Line 92 "in vitro" – Italic. Response 4: Thank you for pointing this out. We agree with this comment. Therefore, We have changed the in vitro format to italics. (line 103) Comments 5: Lines 178-179 About digestion, why did you not choose INFOGEST method? Response 5: Many thanks for your comment. Both Englyst and INFOGEST are classical models of in vitro digestion. INFOGEST is more suitable for the changes, interactions and effects on digestibility of food components in different stages of digestion. Therefore, we chose Englyst, and we will also use INFOGEST in subsequent experiments to determine the more detailed situation of starch digestion. Comments 6: Line 198 "All animal experi ments" Please, correct the word experiments. Response 6: Many thanks for your comment. We have made changes. (line 209) Comments 7: Line 213. "Gene sequencing and data analysis were conducted following a previously described method (Huang et al., 2022)". Please, describe it breafly. Response 7: Many thanks for your comment. We have described the sequencing process in detail. (lines 219-233) Comments 8: Table 5 Please, include in the title the meaning for RDS, SDS and RS. Response 8: Many thanks for your comment. We have explained SDS, RDS, and RS. (line 408) Comments 9: Line 410 "Lactobacillus". Italic Response 9: Many thanks for your comment. We have changed the Lactobacillus format to italics. (line 443) Comments 10: Lines 419-420 and all over the manuscript "Arthromitus, Blautia, norank_f_Desulfovibrionaceae, Colidextribacter, 419 Muribaculum, Eubacterium, Lachnospiraceae, Bifidobacterium, Erysipelotrichaceae, and Desul-420 fovibrio (Figure 4G)." Family taxonimic level (ex. Lachnospiraceae, Erysipelotrichaceae) is not written in italic. Please correct change for italics to not italic. Response 10: Many thanks for your comment. We have changed. (lines 452-454) Comments 11: Line 429 "Relevant studies have confirmed that Clostridia plays a role in disease prevention, diagnosis, and treatment." This statmente is confuse. Can you better explain how Clostria can be relevant to diagnosis and treament? Response 11: Many thanks for your comment. We have elaborated in detail in lines 462-465. Comments 12: Figure 4 - On letter D, there is not only Genus level expressed, as stated in axis y. Please, correct it. Response 12: Many thanks for your comment. We have changed. Comments 13: Figure 4 - Letter C, E, F and G. PM is Polygonum multiflorum Thunb. PMP is processed Polygonum multiflorum What is K? Response 13: Many thanks for your comment. K is Blank control group. Comments 14: Lines 483-484 "Clostridia, Lachnospiraceae, and Lachnospirales species, which play a role in promoting gut health." An increase in Clostridia is not always a good outcome. Many pathogens belong to the genus Clostridium, such as Clostridium butulinum, Clostridium perfrigens and Clostridium difficile. Response 14: Many thanks for your comment. Clostridia are a group of bacteria and cannot be simply classified as either beneficial or harmful bacteria, because this genus contains a variety of different species, and their impacts on the host and the environment vary. Some of them are beneficial. Promoting intestinal health: Some Clostridia can help maintain the balance of the intestinal microbial community. By competing with other harmful bacteria for nutrients and living space, they can inhibit the growth of harmful bacteria. For example, certain strains within the genus Clostridium are able to produce short-chain fatty acids, such as butyric acid, which provides energy for intestinal epithelial cells. This helps maintain the integrity of the intestinal mucosa and enhances the function of the intestinal barrier, thus having a positive impact on overall intestinal health. Participating in substance metabolism: Clostridia can participate in a variety of metabolic processes within the human intestine. They are helpful in breaking down some complex carbohydrates and dietary fibers that are difficult for the human body to digest on its own, converting them into nutrients that can be absorbed and utilized by the human body, thereby promoting human metabolism. In addition, Clostridia also play an important role in the soil environment. They are involved in the decomposition of organic matter and nutrient cycling, contributing to the maintenance of soil fertility and the balance of the ecosystem. We mainly analyze them as beneficial bacteria, and we will conduct a more detailed exploration of them in the follow-up. |
||

Reviewer 2 Report
Comments and Suggestions for Authors
2.12. Animal experimental design: more details is needed in this section.
- Authors should mention where animal were acquired.
- Authors should also mention the IACUC or relevant ethical approval.
- Housing condition of the animals should also be described.
- Numbers of mice in each group should also be explicitly described.
- The volume of the gavaging solution is not mentioned, only providing concentration is not enough for replication.
2.13. Gut microbiota analysis
- How gene were sequenced and analyzed were not described here. (Huang et al., 2022) reference does not exist in the manuscript.
- Line 369: it should be Rarefaction Curves instead of dilution curve.
- More detailed description of Figure 4 legend is needed.
- Figure A and B should be grouped by condition, now it is impossible to tell which group has higher alpha diversity.
- Figure 4F is also very blurry.
- OTU name in 4G is not readable.
- Additionally, the authors should conduct beta diversity analysis with PCoA plot, that can better differentiate among 3 groups.
- In the supplementary material, the authors should provide the detailed OTU map results from the microbiome analysis.
Author Response
Please see the attachment.
Comparison of the Structure, Physicochemical Properties, and Impact on Intestinal Flora of Processed and Unprocessed Polygonum multiflorum Starch
Response to Reviewer 2 Comments
- Summary
Thank you very much for taking time out of your busy schedule to review this manuscript. We have made the modifications as per your suggestions, and all the modified parts have been highlighted in yellow. Thank you again for your suggestion.
- 2. Point-by-point response to Comments and Suggestions for Authors
Comments 1: Authors should mention where animal were acquired.
Response 1: Thanks for your comment. We have fully supplemented the animal experiments. (lines 204-217)
Comments 2: Authors should also mention the IACUC or relevant ethical approval.
Response 2: Thanks for your comment. We have fully supplemented the animal experiments. (lines 204-217)
Comments 3: Housing condition of the animals should also be described.
Response 3: Thanks for your comment. We have fully supplemented the animal experiments. (lines 204-217)
Comments 4: Numbers of mice in each group should also be explicitly described.
Response 4: Thanks for your comment. We have fully supplemented the animal experiments. (lines 204-217)
Comments 5: The volume of the gavaging solution is not mentioned, only providing concentration is not enough for replication.
Response 5: Thanks for your comment. We have fully supplemented the animal experiments. (lines 204-217)
Comments 6: How gene were sequenced and analyzed were not described here. (Huang et al., 2022) reference does not exist in the manuscript.
Response 6: Thanks for your comment. We have described the process in detail, citing the correct literature. (lines 219-233)
Comments 7: Line 369: it should be Rarefaction Curves instead of dilution curve.
Response 7: Thanks for your comment. We have corrected it. (line 411, 416)
Comments 8: More detailed description of Figure 4 legend is needed.
Response 8: Thanks for your comment. We have described Figure 4 legend in detail.
Comments 9: Figure A and B should be grouped by condition, now it is impossible to tell which group has higher alpha diversity.
Response 7: Thanks for your comment. We used the rarefaction curve combined with the diversity index to evaluate whether the sequencing volume this time is sufficient to cover all the microorganisms in the samples. This can directly reflect the rationality of the amount of sequencing data. When the curve tends to be flat, it indicates that the amount of data is reasonable. We further used Beta diversity to reflect the differences among the various groups you proposed, just as you mentioned in your 12th comment.
Comments 10: Figure 4F is also very blurry.
Response 7: Thanks for your comment. We have changed the Figure 4F. (Figure 5A)
Comments 11: OTU name in 4G is not readable.
Response 7: Thanks for your comment. We have changed the Figure 4G to be readable by name. (Figure 5B)
Comments 12: Additionally, the authors should conduct beta diversity analysis with PCoA plot, that can better differentiate among 3 groups.
Response 7: Thanks for your comment. We have supplemented the PcoA diagram. (Figure 4C)
Comments 13: In the supplementary material, the authors should provide the detailed OTU map results from the microbiome analysis.
Response 7: Thanks for your comment. We have filled in the details, as shown in Table S1.
Table S1 the detailed OTU map results from the microbiome analysis.
|
Species name |
K-Mean(%) |
K-Sd(%) |
PM-Mean(%) |
PM-Sd(%) |
PMP-Mean(%) |
PMP-Sd(%) |
P_value |
P_adjust |
|
s__Candidatus_Arthromitus_sp._SFB-mouse-Japan |
0.921 |
0.568 |
0.365 |
0.753 |
0.669 |
1.189 |
0.016 |
0.280 |
|
s__uncultured_bacterium_g__Blautia |
1.071 |
1.710 |
0.018 |
0.078 |
0.358 |
0.529 |
0.000 |
0.004 |
|
s__uncultured_bacterium_g__norank_f__Desulfovibrionaceae |
0.599 |
0.135 |
0.282 |
0.185 |
0.344 |
0.325 |
0.025 |
0.342 |
|
s__uncultured_bacterium_g__Colidextribacter |
0.560 |
0.288 |
0.260 |
0.214 |
0.326 |
0.203 |
0.024 |
0.342 |
|
s__uncultured_bacterium_g__Muribaculum |
0.260 |
0.194 |
0.460 |
0.428 |
0.231 |
0.210 |
0.005 |
0.125 |
|
s__uncultured_bacterium_g__Eubacterium_siraeum_group |
0.461 |
0.466 |
0.113 |
0.217 |
0.059 |
0.118 |
0.041 |
0.390 |
|
s__Lachnospiraceae_bacterium_COE1 |
0.329 |
0.332 |
0.070 |
0.104 |
0.043 |
0.065 |
0.013 |
0.242 |
|
s__Bifidobacterium_pseudolongum |
0.002 |
0.004 |
0.292 |
0.646 |
0.094 |
0.247 |
0.047 |
0.423 |
|
s__uncultured_bacterium_g__norank_f__Erysipelotrichaceae |
0.089 |
0.048 |
0.106 |
0.092 |
0.170 |
0.146 |
0.035 |
0.390 |
|
s__Desulfovibrio_fairfieldensis |
0.235 |
0.261 |
0.047 |
0.065 |
0.082 |
0.117 |
0.039 |
0.390 |

Reviewer 3 Report
Comments and Suggestions for Authors
This study provides a comprehensive analysis and comparison of the structural and physicochemical properties of starch extracted from Polygonum multiflorum before and after thermal processing. Notable structural alterations were identified between raw (PM) and processed (PMP) starch, primarily due to gelatinization effects induced by heat treatment. These modifications significantly influenced starch behavior and characteristics. One of the key findings was an increase in resistant starch content in PMP starch, which is largely linked to changes in gut microbiota composition—specifically, the enrichment of Clostridia, Lachnospiraceae, and Lachnospirales species known for supporting intestinal health.
The study highlights the relationship between processing techniques and starch functionality, offering a multi-layered approach to optimizing traditional Chinese medicine-based functional foods. By fine-tuning thermal processing parameters, targeted adjustments to starch solubility and swelling properties can be achieved, establishing a strategic foundation for material design in food engineering. Furthermore, the observed increase in resistant starch enables the creation of carbohydrate matrices with potential controlled-release behavior.
In addition, the specific modulation of gut flora by PMP starch opens the door for its integration with probiotics, facilitating the development of synbiotic food products aimed at regulating the gut-brain axis. Future research should focus on enhancing the structural-function stability of these drug-food matrices and improving consumer sensory acceptance, thereby enabling the meaningful integration of traditional medicinal insights with contemporary food science.
Author Response
Please see the attachment.
Comparison of the Structure, Physicochemical Properties, and Impact on Intestinal Flora of Processed and Unprocessed Polygonum multiflorum Starch
Response to Reviewer 3 Comments
- Summary
Thank you very much for taking the time to review this manuscript.

Reviewer 4 Report
Comments and Suggestions for Authors
The manuscript addresses a timely and relevant topic by exploring the structural and functional differences between raw and processed Polygonum multiflorum (PM) starch. The work is thorough, and the authors demonstrate commendable effort in their multi-level analysis, from microstructural characterization to in vitro digestion and gut microbiota impact. However, there are several areas where the manuscript could be improved to enhance clarity, scientific rigor, and impact:
- The novelty of studying starch rather than polysaccharides or other constituents in Polygonum multiflorum is acknowledged. However, more context is needed in the Introduction (Lines 67–82) regarding:
- Why starch, as opposed to other known bioactives in PM, deserves focused exploration.
- How the findings can specifically influence functional food or clinical TCM applications
- Methodology – Statistical Rigor and Animal Study Design (Lines 192–206):
- The animal study design is sound, but dividing 65 mice into 13 groups (n=5) weakens statistical power.
- Consider reducing the number of experimental groups or justifying the selection of PMP1-3 and PM1-3 groups.
- More detail on how microbial sequencing data was analyzed (e.g., diversity indices, normalization methods) would benefit the methods section.
- Line 35–38: The description of gelatinization is accurate but verbose. Consider shortening or restructuring for clarity.
- Line 66: “Offers potential health benefits” is vague—specify the health domains (e.g., metabolic, digestive).
- Line 148–153: In the XRD analysis, add representative peak angles (e.g., 2θ values) to support structural assertions.
- Line 83: Revise the sentence: “the structures and physicochemical properties particularly the impact on intestinal flora…” for grammatical clarity. Suggest: “…the structures and physicochemical properties, particularly their impact on intestinal flora…”
- Add a graphical abstract or visual summary to highlight processing effects on starch structure and function.
- The discussion of resistant starch and its correlation with Clostridia/Lachnospiraceae enrichment is promising, but the mechanistic linkage between starch structure and microbial selection remains somewhat implied rather than explained. I suggest incorporating more context or citations regarding how RS promotes butyrate-producing taxa.
- The conclusion could benefit from more direct summary statements of key findings and their significance (e.g., “Thermal processing increased RS content by X% and enriched beneficial gut microbes…”).
- The final paragraph presents exciting applications (synbiotics, controlled-release matrices), but these are speculative. Consider framing them as future directions more explicitly.
Author Response
Please see the attachment.
Comparison of the Structure, Physicochemical Properties, and Impact on Intestinal Flora of Processed and Unprocessed Polygonum multiflorum Starch
Response to Reviewer 4 Comments
- Summary
Thank you very much for taking time out of your busy schedule to review this manuscript. We have made the modifications as per your suggestions, and all the modified parts have been highlighted in yellow. Thank you again for your suggestion.
- 2. Point-by-point response to Comments and Suggestions for Authors
Comments 1: The novelty of studying starch rather than polysaccharides or other constituents in Polygonum multiflorum is acknowledged. However, more context is needed in the Introduction (Lines 67–82) regarding:
Why starch, as opposed to other known bioactives in PM, deserves focused exploration.
How the findings can specifically influence functional food or clinical TCM applications.
Response 1: We greatly appreciate your recognition of the novelty of the research direction regarding the exploration of starch in Polygonum multiflorum in our study. We have incorporated the suggestions you put forward into the original text. (lines 85-94)
Comments 2: Methodology – Statistical Rigor and Animal Study Design (Lines 192–206):
The animal study design is sound, but dividing 65 mice into 13 groups (n=5) weakens statistical power.
Consider reducing the number of experimental groups or justifying the selection of PMP1-3 and PM1-3 groups.
More detail on how microbial sequencing data was analyzed (e.g., diversity indices, normalization methods) would benefit the methods section.
Response 2: Thanks for your comment. Our Experimental Animal Management Center requires that there should be no more than 5 animals in each cage. However, we set up two cages for each group, that is, n = 10 for each group. We didn't write this clearly before and have made the modification. In addition, we carried out three replicates for both PM and PMP respectively to ensure the reliability and validity of the experiment. (lines 213-214)
The details of the sequencing method have been supplemented. (lines 219-233)
Comments 3: Line 35–38: The description of gelatinization is accurate but verbose. Consider shortening or restructuring for clarity.
Response 3: Thanks for your comment. We have changed it to "Gelatinization is a process that starch transforms from ordered semicrystalline granules to an amorphous state." (line 37-38)
Comments 4: Line 66: “Offers potential health benefits” is vague—specify the health domains (e.g., metabolic, digestive).
Response 4: Thanks for your comment. We have given a detailed description of the original text. (lines 65-88)
Comments 5: Line 148–153: In the XRD analysis, add representative peak angles (e.g., 2θ values) to support structural assertions.
Response 5: Thanks for your comment. This part only describes the range of the scanning angle when we conducted the XRD (X-ray diffraction) test. When we discussed the process of observing the XRD patterns later in the text, we emphasized the key angles for that observation.
Comments 6: Line 83: Revise the sentence: “the structures and physicochemical properties particularly the impact on intestinal flora…” for grammatical clarity. Suggest: “…the structures and physicochemical properties, particularly their impact on intestinal flora…”
Response 6: Thanks for your comment. We have changed it to “Herein, the structures and physicochemical properties, particularly the impact on intestinal flora of raw PM starch and processed Polygonum multiflorum (PMP) starch were systematically characterized and compared.” (lines 15-17)
Comments 7: Add a graphical abstract or visual summary to highlight processing effects on starch structure and function.
Response 7: Thanks for your comment. We added a graphic summary. (line 530)
Comments 8: The discussion of resistant starch and its correlation with Clostridia/Lachnospiraceae enrichment is promising, but the mechanistic linkage between starch structure and microbial selection remains somewhat implied rather than explained. I suggest incorporating more context or citations regarding how RS promotes butyrate-producing taxa.
Response 8: Thanks for your comment. We have provided detailed explanations in the original text. (lines 513-523)
Comments 9: The conclusion could benefit from more direct summary statements of key findings and their significance (e.g., “Thermal processing increased RS content by X% and enriched beneficial gut microbes…”).
Response 9: Thanks for your comment. We have made modifications at the original site. (lines 534-536)
Comments 10: The final paragraph presents exciting applications (synbiotics, controlled-release matrices), but these are speculative. Consider framing them as future directions more explicitly.
Response 10: Thanks for your comment. We have made modifications at the original site. (lines 542-545)

Round 2
Reviewer 2 Report
Comments and Suggestions for Authors
The authors had made decent efforts to revise the manuscript.